# Raman Spectroscopy as a Neuromonitoring Tool in Traumatic Brain Injury: A Systematic Review and Clinical Perspectives

**DOI:** 10.3390/cells11071227

**Published:** 2022-04-05

**Authors:** Andrew R. Stevens, Clarissa A. Stickland, Georgia Harris, Zubair Ahmed, Pola Goldberg Oppenheimer, Antonio Belli, David J. Davies

**Affiliations:** 1Neuroscience, Trauma and Ophthalmology, Institute of Inflammation and Ageing, University of Birmingham, Birmingham B15 2TT, UK; z.ahmed.1@bham.ac.uk (Z.A.); a.belli@bham.ac.uk (A.B.); daviesdj@doctors.org.uk (D.J.D.); 2NIHR Surgical Reconstruction and Microbiology Research Centre, University Hospitals Birmingham, Birmingham B15 2TH, UK; 3School of Chemical Engineering, University of Birmingham, Birmingham B15 2TT, UK; cas784@student.bham.ac.uk (C.A.S.); gxh975@student.bham.ac.uk (G.H.); p.goldbergoppenheimer@bham.ac.uk (P.G.O.); 4Centre for Trauma Science Research, University of Birmingham, Birmingham B15 2TT, UK

**Keywords:** traumatic brain injury, neuromonitoring, Raman spectroscopy, point-of-care diagnostics, neuroinflammation, oxidative stress, metabolic dysfunction

## Abstract

Traumatic brain injury (TBI) is a significant global health problem, for which no disease-modifying therapeutics are currently available to improve survival and outcomes. Current neuromonitoring modalities are unable to reflect the complex and changing pathophysiological processes of the acute changes that occur after TBI. Raman spectroscopy (RS) is a powerful, label-free, optical tool which can provide detailed biochemical data in vivo. A systematic review of the literature is presented of available evidence for the use of RS in TBI. Seven research studies met the inclusion/exclusion criteria with all studies being performed in pre-clinical models. None of the studies reported the in vivo application of RS, with spectral acquisition performed ex vivo and one performed in vitro. Four further studies were included that related to the use of RS in analogous brain injury models, and a further five utilised RS in ex vivo biofluid studies for diagnosis or monitoring of TBI. RS is identified as a potential means to identify injury severity and metabolic dysfunction which may hold translational value. In relation to the available evidence, the translational potentials and barriers are discussed. This systematic review supports the further translational development of RS in TBI to fully ascertain its potential for enhancing patient care.

## 1. Introduction

Traumatic brain injury (TBI) is a significant global health problem [1,2]. The incidence of TBI in Europe is estimated at 1012 cases per 100,000 people per year, and 939 per 100,000 globally [3]. The total annual cost globally is estimated in the region of EUR 55 billion [4]. At present, there are no disease-modifying treatments shown to improve outcomes [5], and it remains a leading cause of death and disability across age groups [6]. After an injury, survivors may develop a range of physical, cognitive, and behavioural symptoms which can have a significant and permanent impact on quality of life [7,8], and TBI is associated with an increased risk of neurodegeneration and dementia [9].

‘Primary brain injury’ is sustained from the trauma itself, whilst ‘secondary brain injury’ occurs after injury due to adverse sequelae resulting in further cell death and dysfunction [10]. In the early stage after injury, the development of cerebral oedema and expansion of surgical mass lesions result in increased intracranial pressure (ICP) and compromise of cerebral blood flow and oxygenation [11]. Contemporary therapeutic paradigms are based on the correction and normalisation of these indices as supportive measures [5,11]. However, TBI has a complex pathophysiology, with a number of mechanisms contributing to the development of secondary injury [12]. Mitochondrial dysfunction [13], metabolic failure [14], excitotoxicity [15], oxidative stress [16], and neuroinflammation [17] are established mediators of ongoing neural injury burden after primary trauma and contribute to poor outcomes to an as-yet unquantified extent.

Current neuromonitoring strategies principally centre on ICP through the introduction of an invasive fibre optic probe into the parenchyma [5,11]. Growing evidence for partial brain tissue oxygen monitoring has increased its implementation in clinical practice [18,19]. Microdialysis of parenchymal tissue is implemented in some centres which provide some metabolic indices [20]. Beyond these modalities, few established means of interrogating the state of neural tissue exist in clinical practice. With the growing complexities of known pathophysiological mechanisms in TBI, there is an increasing need for neuromonitoring to increase in breadth, fidelity, and detail, particularly in monitoring tissue biochemistry.

Raman spectroscopy (RS) is a molecular monitoring technique, dependent on the inelastic scattering of photons detectable after excitation from a monochromatic light source. RS has gained interest in medical applications as a sensitive, label-free optical monitoring modality, capable of detecting molecular changes in real time [21,22,23]. The key to the suitability of RS is its non-destructive nature: it permits accurate tissue interrogation at a molecular level without compromising tissue integrity. RS has in recent years developed a growing interest in a number of potential medical diagnostic applications [22]. These include the diagnosis of atherosclerotic plaques during endovascular cardiac interventions, diagnosis of inflammatory bowel disease during endoscopy, and cancer diagnosis in a number of organs and tissue types.

RS is the technique in which the inelastically scattered light detectable from an excited molecule is exploited for information on the molecule’s structure. Upon excitation by a monochromatic light source (e.g., a laser), electrons in a molecule are excited, inducing a dipole. The dipole interacts with the light and can hereafter be distinctly categorised as follows: elastic Rayleigh scattering and inelastic Stokes and anti-Stokes scattering. The Raman effect is observed through the latter two, in which the scattered light has either a higher or lower frequency than that of the laser. For the light to have a higher frequency, as in anti-Stokes scattering, the molecule will have been in an excited state prior to being exposed to light. As the molecule relaxes to its ground state, the emitted photon has more energy in comparison to the photon emitted from the laser. Most molecules, however, are found in their ground state at room temperature, making this event unlikely. For this reason, usually, only Stokes scattering is considered. Stokes scattering consists of a ground state molecule being raised to a higher energy state as a result of exposure to the laser. The molecule returns to an energy state above the ground state: as such, the photon has a lower energy value than that of the laser. RS data acquired are expressed as a function of the Raman wavenumber, the shift being the difference in frequency between the induced light and the scattered light. Each Raman shift is distinct due to the specific molecular vibration each molecule will have, dependent on the number of atoms present, bond lengths, atoms present, etc. The differences recorded provide each molecule a unique, label-free fingerprint, regardless of the concentration of sample present, and is non-destructive, making RS a powerful tool [24] and compatible with use in vivo.

The downfall of RS lies in its inherently weak nature. The number of photons inelastically scattered is comparatively small, compared with the elastically scattered light. This means the signal is easily overwhelmed by fluorescence, especially from organic molecules. To counter this, the lasers used tend to lie in the near-infrared range, the industrial standard being 785 nm, which allows for the least fluorescence without compromising on the resolution. Other approaches to improve outputs have been to take advantage of nanoscale features of surfaces such as gold or copper to chemically and electromagnetically enhance the sample. This is known as surface-enhanced Raman scattering (SERS) and is considered the most effective way to increase the Raman signal. The need to place the sample on a specialised surface for the SERS procedure, however, limits its use to ex vivo applications.

RS has been the subject of growing interest in pre-clinical studies for the identification and study of a number of pathophysiological mechanisms. For example, RS has been used to detect oxidative stress-induced inflammation in human endothelial cells [25], excitotoxicity in epithelial cells [26], and tracking mitochondrial function [27]. Raman techniques have been utilised in alternative brain injury models, demonstrating consistent RS spectral changes in response to injury, including rat ischaemia–reperfusion injury [28], biochemical change in post-traumatic stress disorder rat models [29], and radiation damage to the brain in mice [30]. RS has been used in the identification of pathognomonic biochemical alterations in a number of neurodegenerative disorders in pre-clinical studies [31] and the identification of neoplasia in cerebral tissue [32]. RS has found applications and interest in neuro-oncological surgery in recent years [33,34,35,36,37,38,39,40,41,42,43]. For rapid intraoperative diagnosis of excised tissue, interest has grown in Raman histology, that is, the use of Raman microscopy with deep convolutional neural networks for rapid and accurate prediction of tissue diagnosis for use in the operating theatre [44]. The application of RS for intraoperative differentiation of normal brain tissue and glioma tissue demonstrates a proof of concept for monitoring applications in the human brain [33], but the exploration of this in the context of TBI has been limited.

The aim of this systematic review was to perform a literature search to identify the purpose of RS in TBI. We discuss the current literature on the utilisation of RS methods in pre-clinical and clinical development for monitoring in TBI and explore the translational challenges and the potential future clinical applications.

## 2. Materials and Methods

A systematic review of the literature was performed to identify relevant articles employing RS analysis in the setting of TBI. A comprehensive search was performed on 29 September 2021 following the methodology of the Cochrane Handbook for Systematic Reviewers and presented in accordance with the Preferred Reporting Items for Systematic Reviews and Meta-Analyses (PRISMA) [45].

Two authors (A.S. and Z.A.) systematically searched the following databases, from their respective inception to 29 September 2021: MEDLINE, EMBASE, Cochrane Central, Scopus, Google Scholar, and Web of Science. Reference lists of pertinent articles on the topic were hand-searched for suitable articles. Reference lists of identified articles for inclusion were also hand-searched for suitable articles. An example search strategy for Medline is given below:Exp Traumatic brain injury;Exp craniocerebral trauma;Keywords 1 or 2;Raman spectroscopy;Exp spectrum analysis, Raman;Keywords 4 or 5;Keywords 3 and 6.

### 2.1. Study Selection

Studies were independently screened for inclusion by two reviewers (A.S. and Z.A.). Initial screening based on title and abstract was performed, followed by full-text screening. Inclusion criteria were defined as utilisation of RS methods (alone or in combination with other methods) for analysis of cerebral tissue from either in vitro or in vivo models of TBI, or clinical use in patients with TBI. Exclusion criteria were articles in which no RS methods were used, or no TBI tissue was utilised.

### 2.2. Data Extraction

Data were extracted from included reports using a piloted form. Data extracted included animal model used, number of replicates, Raman techniques used, the wavelength of the excitatory laser, any tissue fixing prior to analysis, reported outcome, analytical methods, spectral signatures identified, and author conclusions.

### 2.3. Risk of Bias

Risk of bias in individual studies was assessed using SYRCLE’s tool for assessing the risk of bias [46], and the overall risk of bias was determined as low, moderate, or high for each included article.

### 2.4. Synthesis of Results

The heterogeneous nature of data collection and analysis methods precludes our ability to combine and synthesise results into a meta-analysis, and therefore, what follows is a qualitative summary of the results.

## 3. Results

### 3.1. Animal Models of TBI

The search identified 131 articles and after the removal of duplicates, 62 remained. Fifteen articles met our inclusion/exclusion criteria and, after full-text reading, seven studies were included in our final analysis (flow diagram in Figure 1). The included study characteristics are summarised in Table 1. Five additional studies utilised RS techniques for ex vivo biofluid assessment in the context of TBI, and four additional studies investigated RS in mechanistic models relevant to TBI. These are discussed separately in this review. Due to the heterogeneity in the data presented, a meta-analysis was not possible; hence, we performed a qualitative analysis of each study (Figure 1).

Banbury et al. [47] utilised a murine model of controlled cortical impact (CCI) to administer either a mild TBI, severe TBI, or no injury (sham). Three days post-injury, the animal was sacrificed. Brain and retina samples were prepared for confocal RS after paraformaldehyde fixation and air drying. 80% of spectral readings were used as input for a ‘self-optimising Kohonen index network’ (SKiNET [48]), a machine learning technique. The remaining 20% were utilised as ‘test’ spectra. With this, RS of the retina was able to ‘predict’ sTBI in 82% of cases and mild TBI in 75.1%. Brain tissue samples demonstrated a number of spectral changes between injury groups. Severe TBI resulted in a modest relative increase at 1447 cm^−1^, accompanied by a significant decrease at 1660 cm^−1^. Mild TBI demonstrated the same decrease at 1660 cm^−1^ but remained unchanged at 1447 cm^−1^ in comparison to sham. Both injury severities demonstrated a decrease in the band at 1266 cm^−1^. Spectra retrieved were compared with a spectral library to ascertain possible biochemical candidates responsible for the spectral shift observed in injury. Cardiolipin and cytochrome c spectra from a library were identified to fit well and as potential candidates for the spectral effect.

Ercole et al. [49] utilised false colour imaging with a Raman microscope in fresh frozen sections of mouse brain tissue 2 and 7 days after severe TBI. Injured specimens demonstrated strong signals at 1003 and 1560 cm^−1^ persisting at 7 days. They also imaged ipsilateral corpus callosum and internal capsule sections from a site distant from the lesion. These demonstrated reductions in peaks at 1301 and 1440 cm^−1^, partially recovering at 7 days. In a similar study by this group, reported by Surmacki et al. [50], they used these time points of 2 and 7 days in a CCI mouse model of TBI. In this study, they took spectroscopic readings from whole-brain specimens from the contralateral brain, ipsilateral brain distant from contusion core, pericontusional tissue, and contusion core tissue, as well as comparison to sham injury. Their results demonstrated changes in the 1440/1660 cm^−1^ ratio both early (2 days) in the contusion core and late (7 days) in contralateral tissue, suggestive of both local and global biochemical changes in lipid/protein ratio content in response to TBI. The further principal component analysis identified changes in the 701/718 cm^−1^ ratio, corresponding with cholesterol/phospholipid. They hypothesised that the relative increase in cholesterol signal in the contusion core may relate to increasing amyloid-β formation. The group also noted striking features of haem protein in the Raman spectra in early samples from contusion core which resolved by day 7, demonstrating the potential of RS for identifying haemorrhagic conversion/resolution.

Mowbray et al. [51] utilised a controlled cortical impact model of TBI from Sprague Dawley rats. The group observed a fall in 1266 and 1660 cm^−1^ peaks in injured specimens when compared with control. Control specimens also had a relatively smaller 1447 cm^−1^ peak in comparison to TBI. The group utilised a handheld probe device (InPhotonics Raman Probe II, Norwood, MA, USA) with a novel transcranial bolt device fitted with a quartz ‘window’ to permit optical access to the parenchyma. The results of spectra gained from ex vivo animal tissue were obtained using this probe system, which demonstrates proof of concept for the translational ability of RS for in vivo human monitoring of brain issue biochemistry. Research by Tay et al. also identified spectral shifts associated with immunohistochemical findings of caspase-3 expression, adding further observations to support a role for RS as a monitoring modality for levels of cell death or survival after injury.

Kawon et al. [52] principally studied the use of Fourier transform infrared microspectroscopy, supplemented by RS to analyse rat brain samples 30 days after focal injury. They too noted RS changes at 1658 cm^−1^, increasing in intensity associated with glial scarring which they postulated to correspond to matricellular protein release by astrocytes. They also observed reductions at 30 days in 2955 cm^−1^ bands which they hypothesised to be linked to reduced levels of lipids from the destruction of healthy tissue.

Hu et al. [53] utilised rat organotypic hippocampal slice cultures for ‘live’ qualitative imaging by stimulated Raman scattering. Slices after cultured were subjected to stretch mechanical injury and cultured with deuterated amino acids and lipids, which were then visualised with stimulated Raman scattering after 24 h. This was able to demonstrate that increased protein and lipid synthesis occurred in injured hippocampal slices in comparison with uninjured tissue. Due to the necessity of deuterated substrates added to culture media, this technique involves labelling. The ‘label-free’ potential of Raman spectroscopic methods is an attractive feature when considering the translational potential to in vivo settings and is not the case with the methods used here. However, this study demonstrates in principle the potential for Raman spectroscopic methods to be able to interrogate the biochemistry of living injured tissue.

### 3.2. Risk of Bias

All of the included studies stated their primary outcomes and described baseline characteristics (Figure 2). However, none of the studies reported whether they performed a sample size calculation, how incomplete data were dealt with, whether outcomes were randomly assessed, blinding of investigators, random housing of animals, whether the allocation of animals to groups was concealed, correct timing of randomisation, and whether a random sequence was generated in terms of study design. Therefore, there is potentially high risk of bias in all of these animal studies. However, future studies may avoid the apparent high risk of bias by following a set of standardised techniques in animal experiments based on the ARRIVE guidelines [54].

### 3.3. Analogous Animal Studies with Applicable Adverse Tissue Conditions

Hackett et al. [55] utilised resonance RS for selective imaging of haem components in the lesion site and perilesional tissue in a collagenase model of intracerebral haemorrhage (ICH) in the rat. Whilst a model of primary haemorrhagic stroke, ICH models can be considered translatable to the TBI pathophysiology given the frequent incidence of contusion, an analogous pathology, in the traumatically injured cohort. This group was able to utilise haem signal intensity to quantitatively identify the haemorrhagic boundary, peri-haemorrhagic tissue, and normal tissue. This was utilised in a multimodal array, with RR, Fourier transform infrared imaging, and X-ray fluorescence imaging, to improve accuracy and quantitative assessment, and was further used to assess haematoma volume response to rehabilitative therapy [56].

Jung et al. [28] investigated RS in ischaemia–reperfusion injury in a rat model. Though an established model, their study had very small sample sizes, with one specimen in each arm. The group was able to gain spectral data from hippocampal (cornu ammonis area 1) tissue post-mortem ex vivo. In ischaemic versus control models, 1276 and 1658 cm^−1^ bands increased in injury, whereas 1300 and 1438 cm^−1^ bands decreased.

Dutta et al. [57] modelled oxidative stress in hippocampal primary network neuronal culture with the addition of hydrogen peroxide in the presence of Fe^2+^. Oxidative stress is an established mechanism of secondary injury after TBI [58]. Based on their results, 725 cm^−1^ and 1320/40 cm^−1^ bands decreased progressively at the measured 20 min intervals over the 80 min post-exposure. This effect of oxidative stress was ameliorated by the addition of ascorbate, which correlated with live–dead cell counts. As such, the group further demonstrated the ability of RS to identify this therapeutic response.

Lakshmi et al. [30] conducted RS studies on radiation damage to the brain tissue in mice. After exposure to 10 Gy irradiation or control (anaesthesia alone), Raman spectra were obtained from both white and grey matter. These demonstrated early shifts in 1440 and 1660 cm^−1^ regions which were sustained over one-week post-exposure.

### 3.4. Biofluid Diagnostics

Detecting biochemical changes via TBI biomarkers using accessible samples such as blood, saliva and urine can provide early information in the vital acute phase (<24 h) of injury incidence [59] for triaging, monitoring, and diagnostics. Highly investigated TBI biomarkers include S100B, glial fibrillary acidic protein (GFAP), neuron-specific enolase (NSE), and ubiquitin C-terminal hydrolase-L1 (UCHL1) [60]. In 2007, the Scandinavian healthcare system introduced S100B into clinical practice guidelines to predict negative CT scans, thus reducing the number of unnecessary scans [61].

A popular modified RS technique is surface-enhanced Raman spectroscopy (SERS) which links the optimum excitation wavelength to the size, shape, composition, and dielectric environment of nanoparticles [62]. This method requires a rough, metal SERS active substrate to absorb the analyte or for the SERS substrate to be coated in nanostructures [63], via metal colloids in a thin film layer. Rickard et al. [64] developed an optofluidic device using SERS capable of detecting picomolar concentrations of *N*-acetylaspartate (NAA) [64]. NAA is an abundant molecule in the central nervous system (CNS) which is released following TBI and is believed to correlate with injury severity and poor prognosis [65]. The device was also able to detect and discriminate concentrations of S100B and GFAP, (glial fibrillary acidic protein) also recognised biomarkers of neurological injury burden [66]. Whilst this device is developed as an ex vivo tool for monitoring the presence of markers in biofluids, its ability to detect such biomarkers in minute concentrations represents the significant potential of RS for diagnostic resolution for biomarker monitoring.

Gao et al. [67,68] utilised SERS with a paper-based lateral flow assay system with a SERS labelled detection antibody to identify S100B in human blood samples. Using blood samples from patients with TBI, they demonstrated a good correlation with laboratory ELISA measured values of S100B concentration in the samples. Their system was developed to offer a reduced cost and time for the identification of biomarkers of TBI. Li et al. [69] similarly implemented SERS for the assessment of neuron-specific enolase (NSE), a biomarker of TBI, in diluted plasma samples from patients with TBI. With a ‘lateral-flow glass-hemostix’-based SERS assay, the group identified a lower limit of detection of 0.74 ng/mL. These studies demonstrate sufficient fidelity and performance of SERS-based assay in clinically relevant biomarker assay to suggest potential translatability to point-of-care testing.

O’Neal et al. [70,71] demonstrated the utility of SERS for the identification of excitatory amino acids in dialysate from a rat model of ischaemic brain injury after middle cerebral artery occlusion. Glutamate and aspartate in microdialysate were found to be distinguishable using SERS. A comparison of a SERS-based assay with high-performance liquid chromatography was able to identify concentrations of glutamate between 0.4 and 5 µmol/L.

### 3.5. Spectral Evidence from Animal Models

Understanding spectral data from RS in TBI is integral to identifying translatable value for clinical applications. Table 2 offers a summary of the literature on observed spectral changes in animal models of TBI and analogous mechanistic animal studies. Clear and reproducible shifts in peaks are identified across multiple studies and models. A notable contradiction is the 1660 cm^−1^ peak: TBI models have identified a consistent reduction in 1660 cm^−1^ intensity in comparison with control [48,51,52,72]. In contrast, an ischaemia–reperfusion model of brain injury identified an increase in the 1660 cm^−1^ intensity. The shift may alter between these pathologies; however, the validity of these results is highly questionable given the use of *n* = 1 specimen in control and test groups.

Metrics offer a means of rationalising an otherwise arbitrary intensity measurement obtained from spectra. Table 3 offers a summary of the literature on ratio-based metrics from TBI animal studies. Identification of a consistent, unchanged ‘reference’ peak which is minimally altered in the context of injury, allows an internal comparator for a dynamic peak of interest. Based on the data, 1440 cm^−1^ is an example of a ‘reference’ peak, with minimal shift in TBI models; 1660 cm^−1^ operates as a ‘dynamic’ peak which alters in injury.

Resolution of early strong band signals from haem occurs between 2 and 7 days [50]. Whilst initial haem signal may obfuscate underlying metabolic tissue signal, identification of alterations in haem signal may be clinically advantageous to identify and monitor the temporal evolution of contusion. Utilisation of RS amongst other methods to delineate haematoma boundaries further suggests the potential for this application [55].

Dysfunctional mitochondrial lipid metabolism plays a role in both acute and chronic sequelae of TBI [99]. Tissue levels of free fatty acids are dynamic following TBI in rat models and may play a role in secondary tissue damage [98]. Differences between lipid concentrations between injured and uninjured brain tissue are also present after injury [100], and lipid composition changes can be identified with RS to discern such changes [50]. Oxidative states of cardiolipin and its interaction with cytochrome c are altered after injury and may be a mechanism for the production of reactive oxygen species [99]. Cardiolipin has been identified as a potential source for band shift after injury [47] and presents a potential for monitoring by RS, though there are consistent strong Raman bands from numerous brain lipids in the 1440 cm^−1^ region [91]. Whilst there is some overlap of lipid and protein signals, increased acquisition of data and utilisation of neural network analysis may provide means to differentiate and utilise such data.

Beyond this, bands in the 2600–3800 cm^−1^ region are able to identify the water content of the brain in the region of 0.75–0.95 to an accuracy of 0.01 [101,102,103]. Real-time assessment of the progression of brain oedema has clear clinical applications for guiding medical and surgical therapy for acute TBI in the intensive care unit setting.

## 4. Discussion

### 4.1. Translational Considerations

RS is a powerful, label-free, non-destructive modality for biochemical analysis. As demonstrated by early studies using ex vivo analysis of injured and normal brain tissue, it can produce relevant biochemical data relatable to tissue integrity, injury burden, and progression or resolution. Similarly, ex vivo techniques have found clinical utility in neuro-oncology settings for ‘operating table-side’ histological diagnostics. Clearly, in contrast to this application, the clinical setting of TBI is directed at maximal tissue salvage, rather than identifying tissue for excision. As such, in vivo RS would be necessary for providing clinically useful data in TBI. Whilst a number of the studies identified to use in vivo modelling of traumatic brain injury, the RS measurements have been taken ex vivo. In vivo applications of RS in TBI are yet to arise, though the technique has found applications in patients in neuro-oncological surgery [33,34,36,37,38,39,40,41,42,43]. Particularly, Jermyn and colleagues report the development of an intraoperative probe used for the detection of brain cancer tissue, compared with healthy brain tissue, and have demonstrated its safe and effective use in humans [34,41,43]. This concept has been further demonstrated with the development of a fibre optic probe for inclusion within a brain biopsy probe, which has been validated in human pilot studies [104]. The success of ‘live-tissue’ RS in vitro amongst included studies, though not in vivo, does further support the concept of gaining biochemical data from living brain tissue using RS.

There are a number of translational barriers to the effective application of RS for the benefit of patients. One consideration is the access to brain parenchyma: unlike near-infrared spectroscopy, the Raman scattering return signal does not display skull penetrance rendering a transcranial approach vulnerable to excessive ‘noise’ from extracranial tissue scatter, to the extent of rendering it impractical. Subsequently, there are three options: (1) intraoperative cortical access; (2) a probe device which allows optical passage across the skull via a ‘window’ [51]; (3) a temporarily implanted fibre optic probe array capable of excitation and detection, either as an intraparenchymal or cortical monitoring device.

### 4.2. Cranial Access

Proof of principle of utilising RS as an intraoperative means of brain tissue interrogation has been established in the field of neuro-oncology [34,41,43]. As described above, numerous publications have described the developments required for the translation of an intraoperative application of RS—specifically, the localised differentiation between tumour tissue and normal parenchyma as a means to optimise maximal safe resection in glioma surgery. This demonstrates in vivo brain RS as a workable concept.

Whilst the intraoperative period provides cortical access opportunity, readings during surgery will be unable to avoid a degree of ambient light. Whilst the highly fluorescent direct lighting could be averted for readings, it is neither practical nor safe to be able to take cortical readings in circumstances at or nearing darkness. RS is a highly sensitive method but vulnerable to such background optical ‘noise’. Adjustments to laser power and background reference measurements can be applied to overcome this potential [40].

Though the intraoperative period provides cortical access for RS monitoring in patients with traumatic brain injury, the majority of the acute phase of severe TBI is spent in the intensive care unit rather than the operating theatre. Single-time-point (intraoperative) biochemical interrogation with RS could provide clinically relevant data which may correlate with the burden of injury and thus prognosis. However, to permit longitudinal data throughout the acute phase, alternative means of access are required.

Our group has explored a quartz lens as a ‘window’ technique as a means for gaining optical access to the intracranial compartment [51]. Mowbray et al. [51] designed and engineered a novel device which functions as a means of eliminating extra- and cranial tissue, with a polished quartz disc as a transparent window to facilitate spectral interrogation of brain tissue intermittently. This carries a relative advantage that the RS is external and has, therefore, no requirement for sterilisation between patients and is reusable within the lifespan of the equipment, with the comparatively simple cranial access device as a single-patient ‘consumable’.

Standard cranial access techniques to facilitate intracranial monitoring devices are well integrated into modern neurosurgical practice. Up to a fifth of patients with traumatic brain injury who require admission to secondary care facilities need such invasive placement of monitoring probes, principally for the monitoring of ICP as a key paradigm in the supportive management after severe injury. In recent years, additional modalities have become integrated, including brain tissue oxygen monitoring and the insertion of microdialysis probes for the sampling of small molecules of metabolic interest (such as glucose, lactate, pyruvate, etc). The standard of care, therefore, involves the invasive component required, and additional monitoring requires only adaptation of the systems already established, with minimal additional risk burden to patients. An example of this principle is the integration of a near-infrared spectroscopy probe with ICP monitoring [105]. This device has utilised existing intracranial access to introduce an optical monitoring modality for the assessment of cerebral haemodynamics, successfully implemented in a small pilot group without safety concerns.

### 4.3. Probe Development

Whilst a cranial window has been demonstrated as a technically feasible means to facilitate cortical RS, it holds a number of potential limitations. The device is too bulky and cumbersome for continual monitoring in clinical practice. Furthermore, obfuscation of the window with blood, inflammatory tissue, serous fluid, or other tissue may render the implanted device unusable for monitoring at depths beyond this layer which would inherently carry its own Raman signal.

An alternative means of facilitating RS monitoring in parenchyma is through the temporary implantation of a probe. This would require a fibre optic system for delivery of excitation wavelengths, and a further fibre optic system for collection and transmission of detected photons to a charge-coupled device (CCD) for spectroscopic analysis. The system at its most basic could comprise two optical fibres, one dedicated to each of these purposes. Filtering and focussing optics, too large for integration at a probe tip, are external to the probe tip and patient. More complex arrays, typically increasing the number of detection fibres or introducing lens tips, have been developed for endoscopic and probe-based monitoring systems [34,41,43]. One difficulty with fibre optic RS is the inherent Raman signal of silica-based fibre optic surrounds, though this can be accounted for with computational methods. Data resolution may be compromised with a reductionist system, but similarly, computational methods have been demonstrated to achieve optimal data interpretation from such data [48].

Implanted cerebral devices are typically for single use. Cost effectiveness of disposable devices should be considered: minimising production costs of consumables and materials required would both increase the likelihood of cost-effective uptake, as well as achieve favourable sustainability in implementation.

### 4.4. Wavelength Selection

Raman signal is weak and is inversely proportional to the fourth power of the excitation wavelength: longer wavelengths result in reduced scatter and, therefore, reduced sensitivity [106]. Conversely, lower wavelengths (particularly in biological tissues) induce autofluorescence: this signal creates significant ‘noise’ alongside the Raman scatter, making achieving usable resolution more challenging. As such, 785 nm is a typically used wavelength as a ‘middle ground’ [22].

### 4.5. Potential Clinical Applications

#### 4.5.1. In Vivo Monitoring: Detailed Biochemical Assessment

Current therapeutic paradigms in acute TBI are aimed at restoring physiological intracranial haemostasis to minimise secondary injury [5]. Most commonly, this involves measurement of ICP via an implanted probe and therapeutic intervention based on avoiding high ICP [11]. However, numerous further known and unknown mechanisms contribute to poor functional outcomes in secondary brain injury after trauma. These include neural apoptosis, glial scarring, axonal degeneration, excitotoxicity, metabolic failure, mitochondrial dysfunction, neuroinflammatory responses, and oxidative stress [12].

Detection of optical biomarkers in the injured brain could have prognostic value and guide further therapy in the acute phase. RS has demonstrated a pre-clinical aptitude for the assessment of aspects of these pathophysiological pathways, as outlined above. For example, metabolic failure from mitochondrial dysfunction is a key factor in poor outcomes from TBI, but the real-time assessment of this is minimally addressed with current standard neuromonitoring and therapeutic methods [107,108,109,110]. In vivo RS monitoring in the gastrocnemius muscle of the rat has been demonstrated to be able to detect early changes in mitochondrial redox states in response to sepsis, prior to any change detected in lactate levels. Translation of this concept to brain tissue as a single example would provide great potential for clinical applications; an illustration of a potential application and a general summary are given in Figure 3.

As described above, there are a number of possible methods of accessing brain tissue for optical interrogation. The standard of care in moderate-to-severe TBI routinely involves the invasive placement of probes into the brain parenchyma and would add a minimal excess risk burden to utilise such access for RS. This would permit ‘online’ or ‘real-time’ detailed biochemical assessment and response to therapy.

Further considerations are required for the utility of Raman spectra in clinical decision making. Given the currently sparse available data, it is not possible a priori to determine the exact influence RS may in the future have on the clinical care of patients with TBI. To consider ICP monitoring in comparison, since first-in-human measurements in 1951, and its integration into standard practice in the 1970s [111], further study and optimisation of the interpretation and clinical applications of the indices and derivatives of ICP are ongoing to this day [112]. However, it can be reasonably argued that a technique such as RS has the potential to offer molecular insights into the function and dysfunction of the human brain far beyond that of intraparenchymal pressure. In discerning the potential utility and clinical relevance of RS in acute traumatic brain injury, clinical studies to gain spectral data from humans with traumatic brain injury would be of tremendous benefit in the exploration of future applications. A summary of commercially available probes which have been applied in neurosurgical settings is available in a review article from DePaoli et al. [113].

The spectral data from implanted fibre optics may be technically problematic. RS is sensitive to a number of factors which may adversely affect data quality and consequently, utility. Implanted fibre optic probes inherently are affected by ‘background’ spectra from silica fibre covering, and using small numerical aperture optics (as is preferable for minimising invasiveness) may further compromise the data. Previously, our research group has used machine learning tools to overcome such difficulties (SKiNET [47,51]). As demonstrated by SKiNET, it is possible to train such a system to accurately discern subtle modalities of biochemical derangement which correspond with differing adverse tissue conditions.

RS is able to identify a spectral ‘signature’ of normalisation of biochemical derangement. This effect has been observed in an oxidative stress model, with normalisation of deranged Raman signal with the addition of ascorbate [57]. Applied in vivo, this permits an immediate optical monitoring mechanism for oxidative stress states. Given the wide-ranging capabilities of RS to resolve spectral fingerprints of biochemical change at the molecular level, there is significant potential for such a concept to be applicable to other pathophysiological mechanisms in TBI. At present, outcome measures in TBI require at least three months of follow-up. Even with extensive longitudinal outcome assessments, the heterogeneity of injury necessitates vast numbers required in clinical trials to demonstrate clinical efficacy. Immediate biochemical feedback of normalisation of such mechanisms which may correspond to neural survival could provide such a proxy measure for early indication of effect.

#### 4.5.2. Microdialysis Online Analysis

A microdialysis is a tool for neuromonitoring of cerebral metabolic function, used in limited centres in clinical practice [20]. Presently, samples of effluent dialysate are collected over intervals (one hour is typical) and intermittently analysed. This creates limitations in both clinical and research practice: changes in metabolite composition of the dialysate are not immediately recognisable, and any change is ‘binned’ into a mean sample for each hour. As such, establishing a true temporal relationship between patient or therapeutic variables and the metabolic observed effect is not possible [114]. RS-based methods have been established as effective in identifying clinically relevant quantitative information on metabolites [70,71]. Given the high sampling rate and accuracy of RS-based methods, there is the potential for continuous analysis of passing dialysate: a bedside system for microdialysis with continuous ‘online’ analysis.

#### 4.5.3. Pitchside/Bedside Concussion Assessment

A number of studies have utilised RS (or a variation thereof) in developing point-of-care applicable diagnostic devices for the recognition of biomarkers of TBI [64,65,66,67,68,69]. Point-of-care detection for TBI has the most relevance in mild TBI and particularly concussion. In severe TBI, the clinical condition of the patient (typified by impaired consciousness or seizure) or the severity of the injury mechanism necessitates urgent radiological diagnosis. However, clinical diagnosis of concussion can be more challenging, both for sportspeople and patients.

Currently, TBI point-of-care diagnostic and monitoring techniques routinely used by healthcare professionals are those of clinical examination and assessment. For moderate-to-severe injury with markedly altered levels of consciousness, the Glasgow Coma Scale (GCS) is the standard examination tool, alongside the assessment of pupillary responses to light. GCS is a fast and simple assessment method that can be performed by a healthcare professional of any level, by assessing the patient’s motor, verbal, and eye responses [115]. Clinical assessment of suspected concussion and mild TBI can rely upon a more nuanced and detailed neurocognitive and symptomatology in addition to an assessment of consciousness, as in the Sports Concussion Assessment Tool, currently in its fifth version (SCAT5) [116].

Clinical examination methods are subjective, with moderate interrater reliability. There are inconsistencies between scores when undertaken by different healthcare workers and at varied timeframes [117]. GCS may also be affected by the presence of intoxicants, medication, sedation, and pre-morbid disability. SCAT5 assessments particularly require patient compliance and participation in both baseline and incident examinations. Attempts to develop objective bedside/pitchside diagnostics in recent years include pupillometry devices to improve interrater reliability of such clinical assessment [118], but as yet, a true point-of-care diagnostic device is not available to clinicians.

The available assessment methods have numerous limitations which highlight the need for an alternative point-of-care method that may be fulfilled by an RS diagnostic device. For maximal clinical utility, prospective diagnostic advice should be relatively portable and rapid for ‘point-of-care’ testing, in both geographical and temporal senses. As with any diagnostic test, the accuracy of the device (specificity and sensitivity) should also be appropriate to the cohort. For a pitch-side application, the ideal device would be user-friendly, such that a monitored first-aid provider would be able to take readings. In this context, the patient would also be outside of a hospital setting, so a pitch-side device would be capable of non-invasive measurements without incisions. RS has also been applied to non-invasive identification of traumatic brain injury diagnosis and severity stratification via examination of the retina [47] and may hold future potential with further development.

## 5. Conclusions

RS is a powerful diagnostic tool with the potential to provide detailed biochemical diagnostic information on the injured brain. This review highlights the success of RS in identifying biochemical changes after injury in tissue from in vivo models and holds significant potential for applications in in vivo or clinical settings for TBI diagnostics and monitoring. Potential applications in TBI are broad and include direct brain monitoring and biofluid diagnostics, with the opportunity to obtain detailed and temporally resolved detail of metabolic and biochemical shifts in response to injury which could increase understanding and inform clinical decision making. Novel spectral biomarkers of injury severity observed from in vivo models represent a significant opportunity for monitoring and treating the complex pathophysiological processes as yet uncaptured by neuromonitoring modalities. Further pre-clinical and clinical studies are required to fully recognise the potential for RS in TBI. Whilst there are translational challenges, discussed examples from neuro-oncology and related optical monitoring modalities set a proof of concept for further investigation and application of RS in TBI.

## Figures and Tables

**Figure 1 cells-11-01227-f001:**
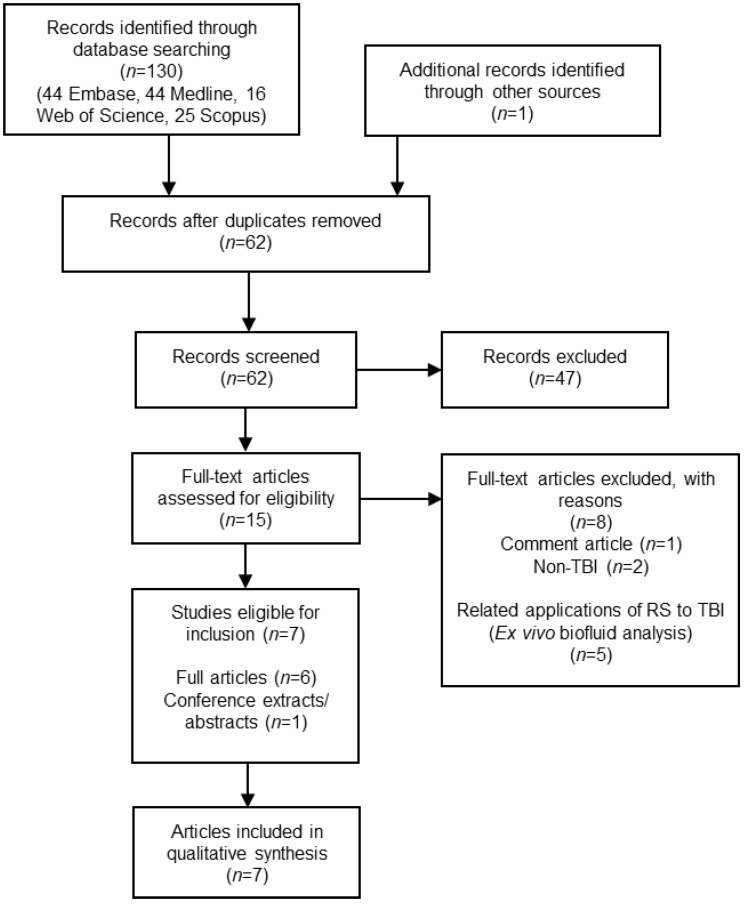
PRISMA flowchart of the systematic review process.

**Figure 2 cells-11-01227-f002:**
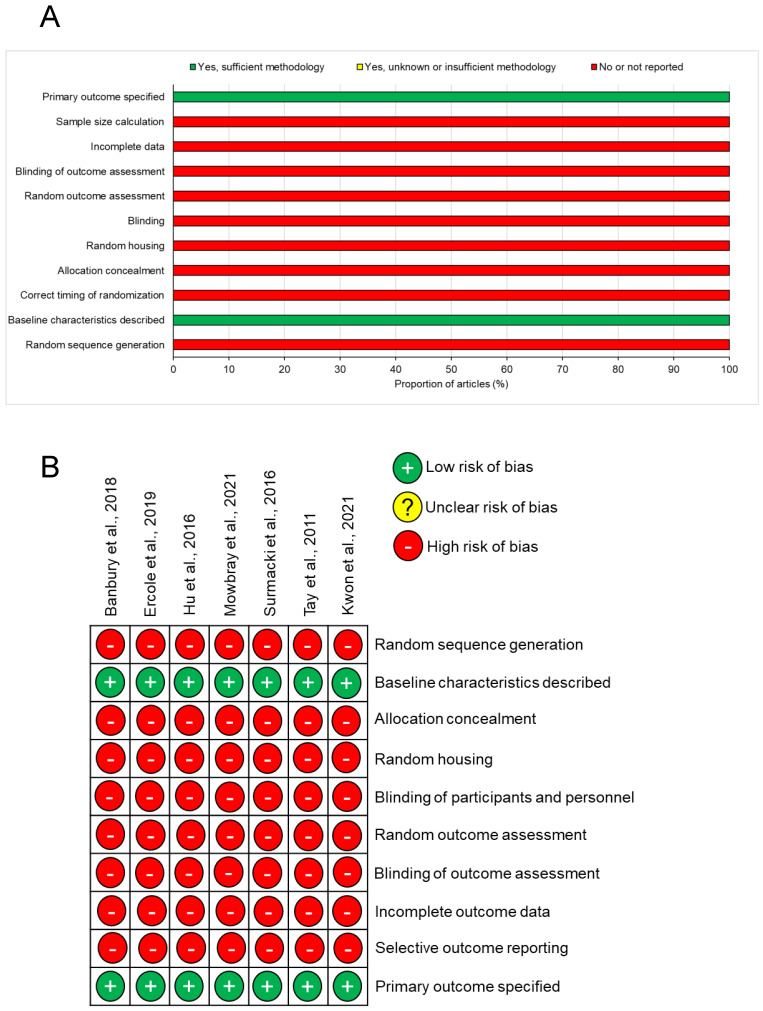
Risk of bias analysis for the included studies using the SYRCLE tool: (**A**) risk of bias summary in all studies; (**B**) risk of bias in individual studies.

**Figure 3 cells-11-01227-f003:**
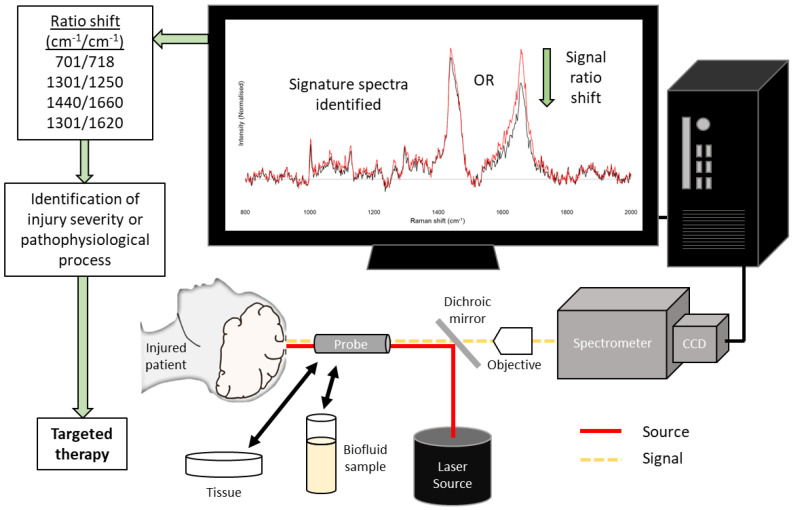
Illustration of a simplified Raman spectroscopy system and its potential application in clinical environments. CCD = charge-coupled device.

**Table 1 cells-11-01227-t001:** Characteristics of the included studies. CCI = controlled cortical impact; FTIR = Fourier transform infrared spectroscopy; mTBI = mild traumatic brain injury; RS = Raman spectroscopy; sTBI = severe traumatic brain injury.

Author	Year	Type	Animal	Model Used	Raman Technique (Wavelength)	Overall Risk of Bias
Banbury et al.	2020	Article	Mouse	In vivo CCI mTBI/sTBI	Confocal RS (785 nm)	Moderate
Ercole et al.	2017	Abstract	Mouse	In vivo CCI sTBI	Confocal RS imaging (785 nm)	Moderate
Hu et al.	2016	Article	Rat	Stretch organotypic hippocampal slice cultures	Stimulated Raman scattering (720–990 nm pump beam, 1064 nm Stokes laser)	Moderate
Mowbray et al.	2021	Article	Rat	In vivo CCI sTBI	Confocal RS (785 nm)	Moderate
Surmacki et al.	2016	Article	Mouse	In vivo CCI	Confocal RS (785 nm)	Moderate
Tay et al.	2011	Article	Mouse	In vivo CCI	Confocal RS (785 nm)	Moderate
Kawon et al.	2021	Article	Rat	In vivo drill focal injury	Confocal RS (532 nm)—Additionally, FTIR-only Raman findings reported in this article	Moderate

**Table 2 cells-11-01227-t002:** Consolidation of Raman spectral data for noted wavenumbers which have demonstrated changes in TBI models or analogous mechanistic models. Assignment data are taken from corresponding references or alternative sources [43,47,50,57,72,73,74,75,76,77,78,79,80,81,82,83,84,85,86,87,88,89,90,91,92,93,94,95,96,97]. Peak assignments based on spectra from excitation with 785 nm laser unless otherwise stated. DNA = deoxyribonucleic acid; Hb = haemoglobin; A = adenine; C = cytosine; G = guanine; mTBI = mild traumatic brain injury; sTBI = severe traumatic brain injury; T = thymine.

Peak Wavenumber (cm^−1^)	Appropriation	TBI
426	Hb [50] Cholesterol [50]	Increasing in contusion core between 2 and 7 days [50]
491-2	Combination modes of the uracil ring plus ribose vibrations [92]	Increasing in contusion core between 2 and 7 days [50]
605	Cholesterol [50]	Increased in contusion core vs. contralateral hemisphere [50]
675	Hb [50], glycerol [93]	Decreasing in contusion core between 2 and 7 days [50]
701-2	Lipids (701 cm^−1^) [50], cholesterol, cholesterol ester	Increasing in contusion core between 2 and 7 days [50]
718	Symmetric and anti-symmetric stretch vibrations of the choline group N+(CH_3_)_3_ in phospholipid, lipids [50]	Decreased in contusion core vs. contralateral hemisphere (2 and 7 days) [50]
725	nucleic acid (A) [57,94]	Reduced oxidative stress [57]
754	Hb [50]	Increased in contusion core vs. contralateral hemisphere (2 and 7 days) [50], decreasing in contusion core between 2 and 7 days [50]
782	DNA peak, cytosine (C), uracil (U), thymine (T), C, T, U-ring breathing [57,95]	Reduced oxidative stress [57]
801	Cyclohexane [74,75,98] (532 nm [96], 575 nm [73] and 785 nm [97] excitation)	Increased in contusion core vs. contralateral hemisphere [50]
850-2	C-H wagging [72] (850 cm^−1^), C−C stretch in proline (collagen) [94]; ring breathing (tyrosine) [94]; C−O−C stretching (glycogen, polysaccharides) [57,94], albumin (852 cm^−1^) [50]	Increased in mTBI and sTBI (3 days) [47]
970	Intralipid [50] haem aggregation marker bands, fibrin components [74]; C-C stretching mixed with C-H rocking, biliverdin, protoporphyrin IX [75]	Decreasing in contusion core between 2 and 7 days [50]
1002	Ring breathing mode of phenylalanine (1002 cm^−1^) [50] and (1005 cm^−1^) [57]	Decreasing in contusion core between 2 and 7 days [50], decreased in sTBI (3 days) [47,51], decreasing in contusion core between 2 and 7 days [50], weak or absent in injured samples [77]
1003	C-C skeletal [72], phenylalanine [76,94], Hb, albumin [50]	Decreasing in contusion core between 2 and 7 days [50], strong signal in pericontusional tissue (days 2 and 7) [49]
1079	Intralipid [50]	Decreasing in contusion core between 2 and 7 days [50]
1090	PO_2_ stretch, phospholipids and nucleic acid [77]	Stable in oxidative stress [57]
1096-8	Phosphodioxy PO_2_ [78], nucleic acid (1097 cm^−1^) [94]	Increased in mTBI and sTBI (3 days) [47]
1154	Phenylalanine, tryptophan, hypro, tyr, phe, m(CC/CN) proteins (1155 cm^−1^) [94] and (1156 cm^−1^) [57]	Decreasing in contusion core between 2 and 7 days [50]
1175	Amide III vibration, cholesterol [77]	Band appears after injury [77]
1224	Hb [50]	Increased in contusion core vs. contralateral hemisphere (2 days) [50], decreasing in contusion core between 2 and 7 days [50]
1227-8	Amide III vibration, phospholipid [77] (1227 cm^−1^); C-H methine bending vibration with changing methaemaglobin content (1228 cm^−1^) [79]	Increased in contusion core vs. contralateral hemisphere [50], band after injury [77]
1266	CH bending modes [72]: amide groups in lipids and proteins [76], cardiolipin [47]	Decreased in mTBI and sTBI (3 days) [47,51], decreased in contusion core vs. contralateral hemisphere (2 and 7 days) [50]
1301	CH_2_ twist/wag/deformation: phospholipid, mixed fatty acid chains, mixed amide III protein vibration [77], lipids [50]	Increasing in contusion core between 2 and 7 days [50], reduction in peak in ipsilateral corpus callosum and internal capsule (distant from lesion) at 2 days, partially recovering by 7 days [49]
1320	CH_2_ CH_3_ twisting; Proteins/lipids nucleic acid [57]	Reduced oxidative stress [57]
1337-1340	Nucleic acid (A,G) [57]; b protein CH_2_ deformation [72], amide III [76] (1337 cm^−1^); collagen (1338 cm^−1^) [94]; albumin(1339 cm^−1^) [50]; CH_2_ scissors, C-OH bending [80]	Reduced in oxidative stress [57], increased in sTBI (3 days) [47,51]
1420	DNA peak, nucleic acid (A, G) [57]; -CH_2_ bending mode for proteins and lipids, indicates change in cytochrome c, intracellular environment, and mitochondrial membrane structures [81]	Reduced oxidative stress [57]
1440	CH_2_ twisting and bending [51,82,96], Lipid, cholesterol, phospholipid, mixed amide I protein, tyrosine [80,94]	Initially decreased in contusion core vs. contralateral hemisphere (2 and 7 days), increasing progressively in contralateral hemisphere and contusion core between 2 and 7 days, to above control levels [50], reduced after ischaemic injury
1447-50	CH_2_ bending [72], mixed proteins and lipids [76] (1447 cm^−1^) [94], albumin, CH_2_ deformation from lipids and proteins (1450 cm^−1^) [77]	Increased in sTBI (3 days) [47,51]
1462	Lipids [50]	Increasing in contralateral hemisphere between 2 and 7 days [50]
1547	Hb [50]	Decreased in contusion core vs. contralateral hemisphere (2 and 7 days), decreasing progressively over time [50]
1560	Associated with mitochondrial activity of cells [77]	Strong signal in pericontusional tissue (days 2 and 7) [49]
1562	Hb [50]	Increased in contusion core vs. contralateral hemisphere (2 days resolving by 7 days) [50]
1576	DNA peak, nucleic acid (A, G) [57]; Hb in red blood cells [82] (though not identified by dedicated erythrocyte RS at 785 nm [83]; N = N stretching vibration [84]	Reduced oxidative stress [57]
1586	Albumin [50]	Sharp band after injury [77]
1618-1620	Predominantly C = O stretch in protein, Hb, Amide I (1618 cm^−1^) [77], tyrosine/tryptophan [94]	Sharp band after injury [77]
1620	Hb [50]	Increased in contusion core vs. contralateral hemisphere (2 days, resolving by 7 days) [50]
1648	Predominantly C = O stretch in protein, Amide I [57]	Decreased in contusion core vs. contralateral hemisphere (2 and 7 days), increasing progressively in contralateral hemisphere over time [50]
1660	Lipids [50], C = C stretching [72,76,82], amide I vibration [57,77], Tyrosine [80,94], mixed lipids and proteins [76], alpha-helix/random coil [94], cardiolipin [47]	Decreased in mTBI and sTBI (3 days) [47,51], decreased in contusion core vs. contralateral hemisphere (2 and 7 days) increasing progressively in contralateral hemisphere and contusion core between 2 and 7 days [50,77]. Increased after ischaemic injury.
1670	Cholesterol, C = C stretching [50]	Marginal increase in contusion core vs. contralateral to lesion [50]

**Table 3 cells-11-01227-t003:** Metrics utilised in the literature for ratio-based comparative analysis of spectral data [48,51,52]. TBI = traumatic brain injury.

Metrics cm^−1^/cm^−1^	Remark	Changes in Animal Models
701/718	Cholesterol/phospholipid	Statistically significant difference between the contralateral and pericontusional/contusional regions at 7 days after TBI [50]
1301/1250	Mixed fatty acid chains/amide III protein	Statistically significant difference between the contralateral and pericontusional/contusional regions at 7 days after TBI [50]
1440/1660	CH deformation/mixed amide I protein and C=C stretching of lipids	Statistically significant difference between the contralateral and pericontusional/contusional regions at 2, 3, and 7 days after TBI [48,51,52]
1301/1620	Mixed amide III protein and fatty acid chains/haemoglobin	Statistically significant difference between the contralateral and pericontusional/contusional regions at 7 days after TBI [50]

## Data Availability

All data generated as part of this study are included in the article.

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
