# Peer review of "Raman Spectroscopy as a Neuromonitoring Tool in Traumatic Brain Injury: A Systematic Review and Clinical Perspectives"

_cells, 2022, doi:10.3390/cells11071227_

Round 1

Reviewer 1 Report

This present study aimed to provide a thorough systematic review and respective clinical perspectives in the application of Raman spectroscopy as a neuromonitoring tool in traumatic brain injury. Your work is a very well documented analysis, together with an extensive list of the corresponding bibliography. Some additional Figures and Tables are summarizing complementary data and interesting comments. In principle, I am appreciating your work and some minor comments, suggestions or corrections are following herewith.

  • I strongly suggest including at the beginning or end of the work a list of abbreviations. In any case, it must be easy, at a given moment, along the read process, to be able to find the correct definition of a given abbreviation.

  • Am I correct that in some cases, you have identified a concrete device, but in other cases, you are introducing or describing methods or studies that have been using devices or laboratory prototypes? Please clarify and indicate if the probe/corresponding device mentioned for the potential clinical application has or not the certification by the FDA (US) or the CE marking (Europe).

Author Response

We thank the reviewer for their kind comments and suggestions for improvement of this manuscript. We address these comments in turn below:

I strongly suggest including at the beginning or end of the work a list of abbreviations. In any case, it must be easy, at a given moment, along the read process, to be able to find the correct definition of a given abbreviation. We have included a list of abbreviations at the end of the manuscript, and ensured that each table and figure contains clarification of each acronym used.

Am I correct that in some cases, you have identified a concrete device, but in other cases, you are introducing or describing methods or studies that have been using devices or laboratory prototypes? Please clarify and indicate if the probe/corresponding device mentioned for the potential clinical application has or not the certification by the FDA (US) or the CE marking (Europe). This is an important point: to our knowledge there is no FDA approved or CE marked device on the market which is approved for use in this context, this review refers to a hypothetical device, based on devices and concepts used in other research applications (mainly in neuro-oncology). We have added a comment on this point with reference to a recent review article which summarises all of the available devices for neurosurgical applications.

Reviewer 2 Report

The review by Ana Margarida Pereira et al, provides an overview of the importance of the use of RS in TBI. The review is clearly written, its original and of interest in its field. I recommend that the review be accepted with minor revision:

  1. The authors should better emphasize the conclusions.
  2. In the introduction section, little previous evidence is provided about the importance of TBI in daily life. Incorporating comparisons with other studies, also preclinical studies, would increase the strength of the paper. Please refer to doi: 10.3390/biology9090238.
  3. The authors should provide an illustrated figure to summarize and simplify reading.
  4. Please provide the full name of all acronyms the first time they are mentioned.
  5. There are some minor grammar issues that should be fixed in order to aid the accessibility of the results to the reader.

Author Response

We thank the reviewer for their kind comments and suggestions. We feel that we now offer a greatly improved manuscript based on the feedback and address these points below.

  1. The authors should better emphasize the conclusions - we have redrafted this section to provide more emphasis
  2. In the introduction section, little previous evidence is provided about the importance of TBI in daily life. Incorporating comparisons with other studies, also preclinical studies, would increase the strength of the paper. Please refer to doi: 10.3390/biology9090238. We have added greater detail on the impact of TBI on patients and quality of life, and added further detail and referencing on the background literature from preclinical studies.
  3. The authors should provide an illustrated figure to summarize and simplify reading We have created a summary figure which we hope provides a simple overview of Raman systems and the proposed applications based on this review
  4. Please provide the full name of all acronyms the first time they are mentioned. We have reviewed the manuscript and have addressed this, based on the suggestion of another reviewer we have added an acronym list at the end of the manuscript
  5. There are some minor grammar issues that should be fixed in order to aid the accessibility of the results to the reader. We have reviewed the manuscript for grammar issues and made minor corrections where identified

Reviewer 3 Report

Dear authors,

Thanks for your manuscript entitled" Raman spectroscopy as a neuromonitoring tool in traumatic brain injury: a systematic review and clinical perspectives by Andrew R. Stevens written well. I read this review with interest and felt it needs some modification for acceptance in this reputed journal.

Comments and suggestions

  1. Abstract should be more scientifically sound. In abstract in some places has written in vivo normal style and some places italic. It should be consistent throughout the manuscript.
  2. Abstract line 25 The literature supports the further translational development of RS in TBI to fully ascertain its potential for enhancing patient care. It will be  Review or literature, please check and confirm
  3. In the introduction section line 73, The molecule returns to an energy state above the ground state: as such, the photon has a lower energy value than that of the laser, why above italic?
  4. Materials and method section  (A.S. and Z.A.)  write consistently.
  5. It's better to mention inclusion and exclusion criteria separately.
  6.  Table 1 includes more related studies and write all abbreviations below of the table that is used in table 1.
  7. Figure 2 a obscure. Increase resolution for better visualization. 
  8. Line 269-274  Surface-Enhanced Raman Spectroscopy (SERS), abbreviation repeated, please check and confirm.
  9.  Table 2 & 3 write all abbreviations below of the table that is used in table 2 & 3
  10. Wavelength selection has been written without any references. 
  11. Line 422, CCD if you used first time abbreviate it.

Author Response

We thank the reviewer for their kind comments and suggestions. We feel that we now offer a greatly improved manuscript based on the feedback and address these points below.

  1. Abstract should be more scientifically sound. In abstract in some places has written in vivo normal style and some places italic. It should be consistent throughout the manuscript. We have made these changes in the abstract and throughout the manuscript
  2. Abstract line 25 The literature supports the further translational development of RS in TBI to fully ascertain its potential for enhancing patient care. It will be  Review or literature, please check and confirm We have changed this to "Systematic Review"
  3. In the introduction section line 73, The molecule returns to an energy state above the ground state: as such, the photon has a lower energy value than that of the laser, why above italic? This was to provide emphasis but we have altered this to normal typeface to avoid potential confusion
  4. Materials and method section  (A.S. and Z.A.)  write consistently. We have reviewed the manuscript and made adjustments for consistency
  5. It's better to mention inclusion and exclusion criteria separately. This has been enacted
  6.  Table 1 includes more related studies and write all abbreviations below of the table that is used in table 1. We have updated the abbreviations list
  7. Figure 2 a obscure. Increase resolution for better visualization. We have provided a higher resolution version
  8. Line 269-274  Surface-Enhanced Raman Spectroscopy (SERS), abbreviation repeated, please check and confirm. This has been corrected
  9.  Table 2 & 3 write all abbreviations below of the table that is used in table 2 & 3 We have updated the abbreviations lists
  10. Wavelength selection has been written without any references. This section has been updated with references
  11. Line 422, CCD if you used first time abbreviate it. This has been corrected